# NeuroMotion smartphone application for remote General Movements Assessment: a feasibility study in Nepal

Antti Juhani Kukka ,[1,2] Heléne E K Sundelin,[3,4] Omkar Basnet,[5] Prajwal Paudel,[6] Kalpana Upadhyay Subedi,[6] Katarina Svensson,[7] Nick Brown ,[1,2] Helena Litorp,[1,8] Rejina Gurung,[1,5] Pratiksha Bhattarai,[5] Johan Wrammert,[1] Ashish KC [1,9]

For numbered affiliations see end of article.

**Correspondence to**
Dr Antti Juhani Kukka;
antti.kukka@uu.se

## ABSTRACT

**Objectives** To evaluate the feasibility of using the NeuroMotion smartphone application for remote General Movements Assessment for screening infants for cerebral palsy in Kathmandu, Nepal.

**Method** Thirty-one term-born infants at risk of cerebral palsy due to birth asphyxia or neonatal seizures were recruited for the follow-up at Paropakar Maternity and Women's Hospital, 1 October 2021 to 7 January 2022. Parents filmed their children at home using the application at 3 months' age and the videos were assessed for technical quality using a standardised form and for fidgety movements by Prechtl's General Movements Assessment. The usability of the application was evaluated through a parental survey.

**Results** Twenty families sent in altogether 46 videos out of which 35 had approved technical quality. Sixteen children had at least one video with approved technical quality. Three infants lacked fidgety movements. The level of agreement between assessors was acceptable (Krippendorf alpha 0.781). Parental answers to the usability survey were in general positive.

**Interpretation** Engaging parents in screening of cerebral palsy with the help of a smartphone-aided remote General Movements Assessment is possible in the urban area of a South Asian lower middle-income country.

## BACKGROUND

All children with disability have the right to reach their full developmental potential.[1] Cerebral palsy (CP) is the most common motor disability in childhood and international guidelines recommend early intervention for all children at high risk of CP.[2] Several ongoing trials are evaluating ways to adapt early intervention to low- and middle-income countries (LMICs),[3–5] which shoulder the vast majority of children affected by CP globally.[6] Delay in accessing rehabilitation is common due to poor availability and lack of targeted screening programmes for high-risk children.[7 8]

Affordable and accurate tools are therefore needed to identify children at high risk of CP

---

## STRENGTHS AND LIMITATIONS OF THIS STUDY

⇒ Three independent assessors judged the presence of General Movements in each video.
⇒ Standardised assessment of the technical quality of videos was performed.
⇒ Social desirability bias might have contributed to the positive results in the usability survey.
⇒ A small sample from a single referral hospital limits generalisability of the findings.

---

in LMICs. General Movements Assessment (GMA) is a cheap, non-invasive method for recognising these children based on observing their spontaneous movements.[9] It has been shown to have excellent validity at 3 months' age and its use in screening high-risk children is now globally endorsed,[10 11] although its accuracy in LMIC populations is yet to be established.[12] The lack of providers trained in the methodology has thus far limited its implementation,[13] and telemedicine is one way of overcoming this barrier.[14 15] Lockdowns imposed during the recent COVID-19 pandemic added urgency for finding alternatives to traditional face-to-face check-ups in neonatal follow-up.

Previous studies in high-income countries (HICs) have shown that remote GMA using smartphone applications (apps) is feasible.[16–18] Automatised movement analysis using machine learning holds further promise for wider availability.[19] However, to our knowledge, no previous study has used smartphone apps for remote GMA in LMIC settings, where the disease burden is the greatest. We therefore aimed to evaluate the feasibility of using smartphone-aided remote GMA for screening of infants at risk of CP in Kathmandu, the capital of Nepal, a lower middle-income country. Feasibility was assessed through the following specific objectives:

1. To estimate the *reach* of the screening programme.
2. To evaluate the *technical quality* of the videos and the *quality of the assessment.*
3. To survey parental perceptions of the *usability* of the app.

## METHODS
A combination of the Strengthening the Reporting of Observational Studies in Epidemiology checklist for cohort studies[20] and Consolidated Standards of Reporting Trials statement for pilot and feasibility trials[21] was used for reporting the study following guidelines for reporting non-randomised pilot and feasibility studies.[22]

### Study design and setting
This is a single-centre feasibility study conducted at the Paropakar Maternity and Women's Hospital, Kathmandu, Nepal. Recruitment of infants ran from 1 October 2021 to 7 January 2022 just before the onset of the third peak of the COVID-19 pandemic and the last response to the usability survey was registered on 2 March 2023. Paropakar is a public tertiary hospital acting as the referral hospital for the surrounding Bagmati province. Infants admitted to the hospital's neonatal intensive care unit/sick newborn care unit (NICU/SNCU) are offered post-discharge follow-up at the paediatric outpatient department up to 14 years age, but few families choose to participate.

Nationally only 70% of infants receive a post-natal check-up in Nepal.[23] Targeted developmental clinics for high-risk children are generally lacking whereby delay in diagnosis until nearly school age is common.[24 25] Paediatric rehabilitation services for CP are provided by non-governmental organisations and government hospitals working by Community-Based Rehabilitation principles.[26] Ownership of smartphones in the Bagmati province is 75 and 84% among women and men, respectively.[23]

### Participants
Term-born neonates (≥37 completed gestational weeks according to last menstrual period) at risk of CP due to birth asphyxia or seizures were recruited for the follow-up. The included infants had to survive to discharge and fulfil at least one of the following criteria: (1) Apgar score ≤5 at 5 min; (2) bag-and-mask ventilation ≥5 min after delivery; or (3) diagnosis of clinical seizures at the NICU/SNCU.

Newborn infants who were transferred to other hospitals, and those whose parents lived outside of the Kathmandu Valley with travel time to the hospital above 1 hour were excluded due to the limited availability of rehabilitation resources outside the capital.

All infants potentially eligible for the study were routinely admitted to the NICU/SNCU, which was confirmed by a chart review of three preceding months. Paediatricians working at the wards identified eligible infants and informed a research assistant in charge of recruitment. Background demographic data and delivery details were collected through a short parental interview and patient chart review, respectively. All participating parents were contacted by phone call when their child was 6 weeks old to enquire about the health of the child and to give parents the possibility to ask further questions about the study.

### Sample size
In preparation for this study, we calculated that to show the sensitivity and specificity of GMA for CP diagnosis at 2 years of age with 95% certainty when the expected sensitivity and specificity are 95% and the expected proportion of infants with abnormal outcomes in the population is 20%, then using Buderer's formula,[27] 365 infants would need to be recruited. A final sample size of 400 would have been needed to account for losses to follow-up. To test the feasibility of recruitment, filming and referral, a convenience sample of 40 neonates was planned for. Based on our previous experience with the hospital, around 2% of deliveries were expected to fill the inclusion criteria.[28]

### Intervention
The NeuroMotion app was translated from Swedish to Nepali and English and adapted to the local context by adding pictures and the possibility of storing videos on the phone for later submission in case the internet was unavailable at the time of filming (figure 1). A trained research assistant guided the parents to install the app on their smartphones. Due to the delay in the app modification, parents of the first 22 infants were instructed to do the installation at home while nine families received the app before discharge from the hospital and two parents without smartphones were invited to come to the hospital to have their child filmed by a research assistant.

Parents were provided with individual login credentials to the app. All parents were instructed to film two videos of their child during the fidgety movements (FM) period 12–16 weeks post-term age.[9] The app automatically sent notifications to the parents when their child was 12 weeks post-term age. The first notification was sent 3 days before scheduled filming, the second on the day of the filming and the third 3 days after the scheduled date if no film had been sent in by then. The research team contacted the parents by phone if no film had been uploaded after the third notification. A second film was requested by a notification in the app 2 days after the first film had been sent or by phone after the film analysis, whichever came first.

The filming took place at home or the hospital by positioning the child in a supine position when awake and satisfied with arms and legs visible on the screen according to the GMA guidelines.[9] The duration of the films was 2–3 min each as in previous studies.[17 18] The app automatically uploaded the films to a secure server at Linköping University, Sweden. Apart from the videos and the pseudonymised study ID, no other data were transferred. All videos were analysed according to Prechtl's GMA by three independent certified evaluators blinded to the medical history of the patient.[9] Results were recorded as present (continuous, intermittent or abnormal), or absent

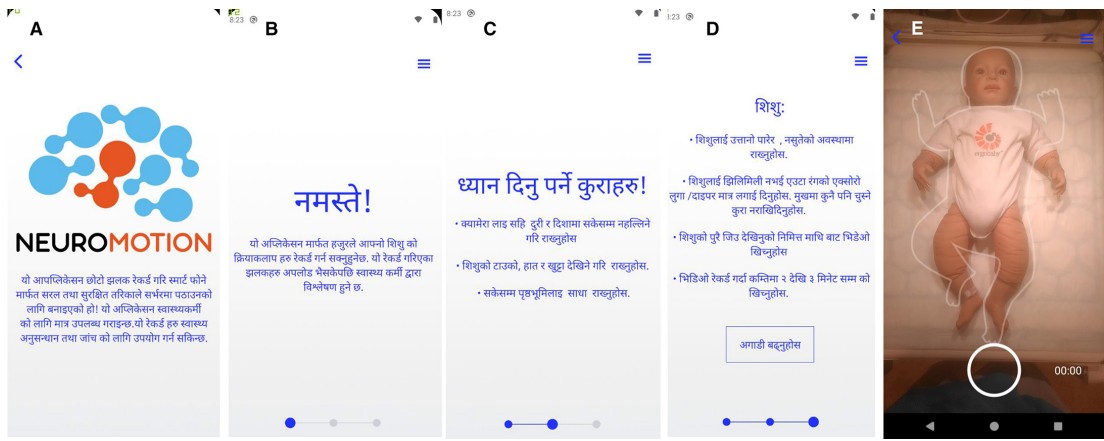

**Figure 1** Screenshots of the Nepali version of the NeuroMotion application including (A) introduction, (B–D) instructions about filming and (E) a view of the filming interface. Corresponding sections in the English version read: (A) this application is designed in order to record and send short clips in a simple and secure way from a smartphone to a secure server only available to healthcare personnel. The clips can be used as support in medical research and diagnostics. (B) Hello! With this application you can record your child. The film can be uploaded and analysed by healthcare personnel. (C) Remember! Keep the camera as still as possible, same distance and direction. Head, hands and feet are visible in the screen. Background as neutral as possible. (D) A baby: should be on its back, be active and satisfied. Only wear a one-coloured t-shirt/one piece and diaper, no pacifier. Film from above, so that the whole child can be seen on the screen. The recording should be 2-3 minutes.

(sporadic or absent) FMs. The assessments were discussed in meetings held every second week. When more than one film was available for a child, the best result category was recorded.

The final consensus agreement was reported to the researchers in Nepal, who in turn informed the families via phone. In case of not approved technical quality or absent FMs, one further recording was requested. Babies with consistently absent FMs were examined by a paediatrician at the Paropakar Maternity and Women's Hospital (PP) and offered a referral for an early intervention programme at Self-help Group for Cerebral Palsy, Lalitpur, after parental consent. The research team offered to cover the costs of the rehabilitation.

After receiving feedback about their child's GMA, parents were interviewed by phone about the usability of the app by applying a multiple-choice questionnaire in Nepali. The survey consisted of 19 questions with answers on a 5-point Likert scale with an option for free-text answers. The questions were based on a survey developed by Kwong *et al* for the evaluation of the Baby Moves app in Australia.[29] Similar questions have also been used in studies in Europe.[17 18]

### Outcomes and statistical analyses

Feasibility was defined as the extent to which smartphone-aided remote screening could be successfully carried out within our setting.[30] The following outcomes were measured quantitatively:

1. *Reach* of the screening programme was evaluated by the proportion of participating children in each stage of the follow-up. Background factors between children to parents who successfully returned at least one film with approved technical quality were compared with those who did not by using independent samples t-test for normally distributed continuous variables and Pearson's $\chi^2$ test or Fischer's exact test for categorical variables depending on the cell count. P value<0.05 was considered statistically significant. Crude ORs with 95% CIs were counted for the categorical variables.

2. The *technical quality* of the videos produced was evaluated by a single evaluator (HEKS) using standardised GMA Trust questionnaire (online supplemental table 1). Each video was assessed by eight parameters. Each parameter was scored as either 'Excellent', 'Good' or 'Indistinct'. The presence of any single parameter with 'Indistinct' score made the video 'Not approved' for technical quality. The *quality of the GMA assessment* was evaluated by calculating inter evaluator agreement with Krippendorf alpha for videos with approved technical quality.[31]

3. Parental perceptions of the *usability* of the app were evaluated by a multiple-choice questionnaire and the results were presented using stacked bar graphs.

Qualitative evaluation on the acceptability of the intervention will be reported separately (submitted manuscript).

### Patient and public involvement

No patient is involved.

### Ethics

Oral and written information was provided to at least one of the parents and written consent was collected prior to enrolment. The research assistant giving the information was hired for the study and was not part of the clinical care team.

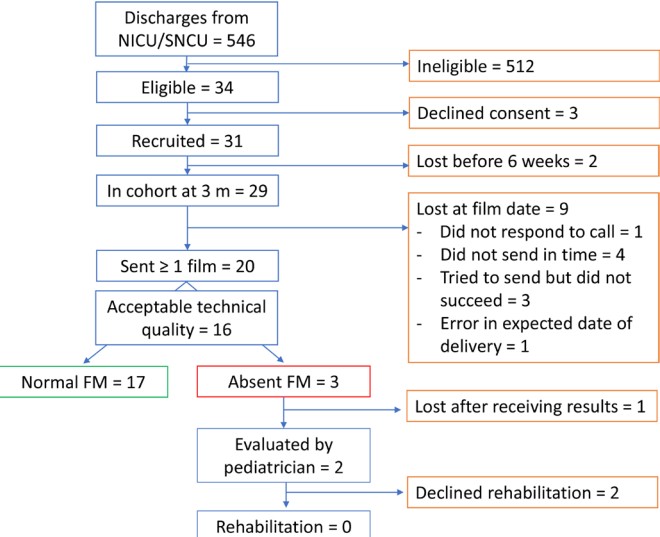

**Figure 2** Participant flow diagram. FM, fidgety movements; SNCU/NICU, sick newborn care unit/neonatal intensive care unit.

## RESULTS
### Reach and quality
The rate of birth asphyxia was lower than expected during the study period and more families were ineligible due to residence outside of the study area. Furthermore, during the first half of the study mortality at the NICU/SNCU was higher than expected and more infants were transferred to other hospitals. Therefore only 34 eligible infants were identified and 31 of them enrolled despite the extension of the recruitment period by a month (figure 2). Male predominance was notable with only six female infants recruited into the study.

Altogether, 46 videos were received from 20 families. All films were made between 12 and 16 weeks of post-term age and thus suitable for GMA analysis. The technical quality was approved in 35 out of 46 videos and 16 children had at least 1 video with approved technical quality (online supplemental table 2). GMA scoring could confidently be performed in four additional videos with not approved technical quality bringing the total number of children assessed by GMA to 18. The most common quality issues leading to the rejection of videos were the inability to hold the camera still (n=15, 33%), the wrong position of the camera for filming (n=12, 26%) and too busy background (n=11, 24%). The level of agreement between the GMA evaluators was acceptable (Krippendorf alpha 0.781 which all three assessors agreed).

There were no statistically significant differences in the characteristics between children whose parents succeeded in returning at least one film of approved technical quality versus those who did not (table 1). Three children had consistently absent FMs. One family moved abroad and could not be reached for follow-up. Two children were evaluated by a paediatrician (PP) and recommended early intervention, but their parents decided against joining the programme.

### Usability
A response to the usability survey was provided by parents of 25 participating children including 18 parents who had succeeded in filming their child and 7 who had attempted to use the app but not succeeded in sending a video (figure 3). Most of the parents found that the app was easy to use, and the instructions were easy to follow. The screen filter outlining the contours of a child helped parents in positioning the camera and sending the videos was easy in most of the cases. More than half of the participants experienced worry about their child's development when using the app and most felt using the app increased their awareness of their child's development. A face-to-face assessment would have been preferable to filming for eight respondents (32%).

## DISCUSSION
This study examined the feasibility of implementing smartphone-aided remote GMA for screening of infants for high-risk CP in Nepal. We found that just short of half of eligible parents (16/34) were able to return at least one film of approved technical quality. Overall, 35 out of 46 films were of approved technical quality. Parental perceptions of the follow-up were positive.

To our knowledge, this is the first time that a smartphone app has been used for assessing GMA outside high-income settings. Our group was more likely to consent to participate (31/34, 91%) than that reported in an Australian cohort of preterm infants (227/273, 83%).[29] However, the proportion of successfully filming was lower in Nepal (16/31, 52%) than in Australia (158/226, 70%),[29] Belgium, Denmark and Norway (69/86, 80%)[17] or Sweden (36/37, 97%).[18] This was mostly due to the technical quality of films being lower than in the previous studies that all reported approved quality in more than 95% of the videos. This contrasts with the findings of the usability survey, where the vast majority of the Nepalese parents reported that they found using the app easy and instructions easy to follow.

Picture sharpness alone rarely was an issue whereby we infer that to improve the success rate of filming, more emphasis should be put on parental training in future studies. In our study, only nine participants received hands-on assistance in installation of the app at the hospital and the rest were guided remotely by phone. A follow-up by phone at 6 weeks' age was added to reduce attrition and a telephone contact was made in most cases at the time of filming. Kwong et al in Australia and a small pilot study in India recently demonstrated that simple instructions for filming could be used to achieve good results with remote GMA where no dedicated app is available.[32 33] However, video file transfer methods would need to comply with local data security and transfer laws.[15]

Parents participating in the study had positive experiences with the NeuroMotion app and the follow-up in general. The response rate to the usability survey was higher than in the previous studies and we also

**Table 1** Background characteristics. Comparison between those participants who returned at least one film of approved technical quality and those who did not

| | Filmed successfully=16 | Not filmed successfully=15 | P value | cOR (95%CI) |
|---|---|---|---|---|
| Infant characteristics | | | | |
| Gestational age in weeks, mean (SD) | 38.9 (1.4) | 38.8 (1.5) | 0.32* | – |
| Apgar at 5 min, mean (SD) | 5.3 (0.4) | 4.8 (0.7) | 0.67* | – |
| Resuscitation at 5 min | 16 | 14 | 0.29† | NA |
| Clinical seizures | 3 | 7 | 0.14‡ | 0.26 (0.05 to 1.32) |
| Sex | | | 1.00‡ | |
| Female | 3 | 3 | | Ref |
| Male | 13 | 12 | | 1.08 (0.18 to 6.42) |
| Mode of delivery | | | 0.25‡ | |
| Normal vaginal | 13 | 9 | | Ref |
| Instrumental | 0 | 0 | | – |
| Caesarean section | 3 | 6 | | 0.34 (0.06 to 1.75) |
| Demographic factors | | | | |
| Maternal education | | | 0.36‡ | |
| 1–7 years | 5 | 3 | | Ref |
| None | 0 | 2 | | – |
| Complete primary | 3 | 2 | | 0.45 (0.05 to 3.57) |
| Complete secondary | 3 | 6 | | 0.30 (0.04 to 2.20) |
| University degree | 5 | 2 | | 1.50 (0.17 to 13.22) |
| Maternal ethnicity | | | 0.90† | |
| Relatively disadvantageous | 11 | 10 | | Ref |
| Relatively advantageous | 5 | 5 | | 1.10 (0.24 to 4.96) |
| Maternal age, mean (SD) | 25 y 5 m (3 y 6 m) | 24 y 2 m (3 y) | 0.60* | |
| Paternal ethnicity | | | 0.90† | |
| Relatively disadvantageous | 11 | 10 | | Ref |
| Relatively advantageous | 5 | 5 | | 1.10 (0.24 to 4.96) |
| Paternal age, mean (SD) | 29 y 6 m (4 y 7 m) | 27 y 7 m (3 y 3 m) | 0.37* | |
| Smartphone use | | | | |
| Parents own a smart phone | 16 | 15 | NA | – |
| Years with smart phone, mean (SD) | 9 y 1 m (3 y 2 m) | 7 y 7 m (3 y) | 0.68* | – |
| Parents use social media | 15 | 15 | 0.33† | NA |
| Application installation | | | 1.00‡ | |
| Hospital | 5 | 4 | 1.00 | Ref |
| Home | 11 | 11 | 1.00 | 0.80 (0.16 to 3.79) |

Parental ethnicity was self-reported. Relatively disadvantageous = Dalit, Janajati, Madhesi or Muslim, relatively advantageous = Brahmini, Chhetri or other.
*Independent samples t-test.
†Pearson's $\chi^2$ test if a cell has count five or more.
‡Fischer's exact text if a cell has count less than 5.
cOR, crude Odds Ratio; m, months; NA, Not available; SD, Standard Deviation; y, years.

interviewed parents who had failed in filming their child as intended. The results were similar to HIC settings with the Nepalese and Australian parents showing higher rates of worry and preference to physical examination than the other two studies.[17 18 29] A separate qualitative study will explore the experiences of the participating parents in-depth (manuscript submitted).

The United Nations' Sustainable Development Goals agenda of leaving no child behind[34] can only be achieved if comprehensive neonatal follow-up is provided

## Usability survey results (n=25)

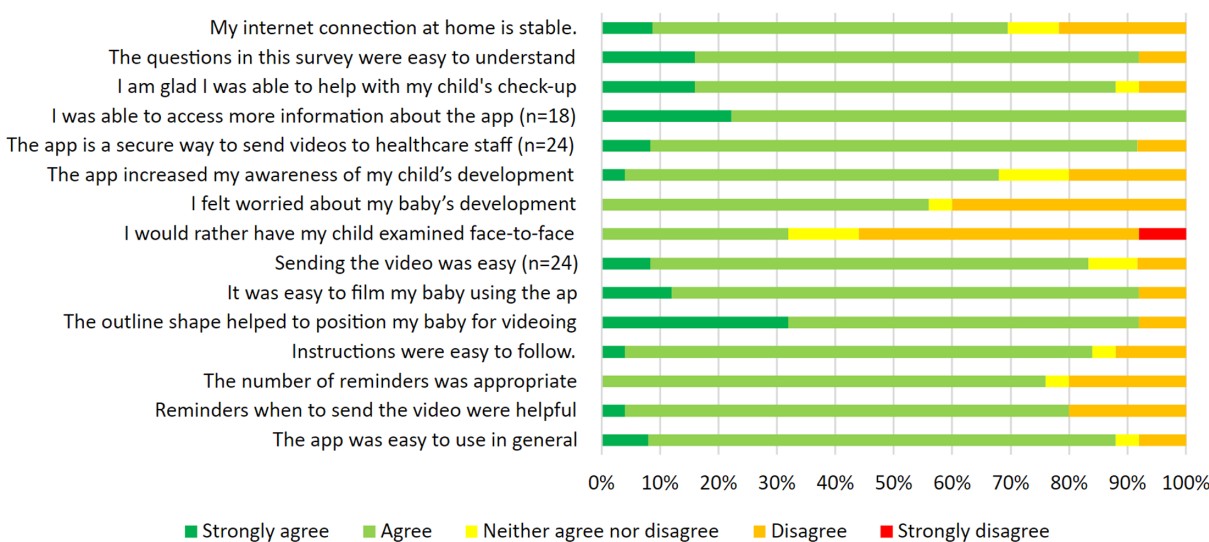

**Figure 3** Stacked bar graph of parental answers to smartphone application usability survey.

simultaneously to improvements in prepartum and peripartum care. The validity of GMA in LMIC settings is yet to be proven as most studies conducted thus far have been small like ours.[12 13] Broad developmental assessment should be provided to children at risk of disability as the GMA has poorer predictive ability for other adverse neurodevelopmental outcomes than CP.[35] A report from Australia published during the COVID-19 pandemic showed that GMA analysis could be made during a Zoom call as a part of a comprehensive developmental evaluation.[36] In LMIC settings, GMA could be added to risk child follow-up by engaging community health workers in filming, which might improve the quality of the films. In India, an ongoing trial is evaluating the possibility of teaching community health workers the GMA method,[3] and in the future automatised analysis could help in the mass screening of videos when filming is done in a standardised manner.[19]

Lastly, ensuring linkage to an early intervention programme is imperative for the ethical conduct of GMA screening. In our study, three children at high risk of CP with absent FMs were identified, but one of the families moved abroad and the other two were not willing to enrol in the offered free intervention programme. The ability of GMA to identify children at high risk of CP before overt signs of the condition appear can be a drawback in settings like Nepal, where there is no tradition for postnatal follow-up and parents have limited resources available. Furthermore, care seeking from alternative providers is common in Nepal as CP is perceived to be caused by also other than medical reasons.[24 37] Further qualitative studies might shed light on the appropriate ways of informing parents about the diagnosis of high-risk CP in low-income and middle-income settings.[38] Early intervention conducted partially via digital platforms might also increase the uptake considering the

overall positive attitudes parents showed towards remote screening.

### Limitations

This study was a small single-centre feasibility study conducted in the capital city of Nepal with a relatively high standard of living, education and mobile phone access and the results are thus not generalisable to rural areas of the country, where remote screening could be most needed. Despite extending the recruitment period we did not reach the planned sample size. The study was not powered to properly examine background predictors that might have impacted filming success and it is essential that alternative means of assessment are provided parallel to remote assessment to avoid discriminating against disadvantaged groups such as women who only represented 20% of the enrolled infants.

We attempted to minimise participation bias by interviewing parents who did not succeed in filming. However, as the interview was conducted by phone by the same research assistant who had provided guidance regarding the app installation and the results of the GMA analysis, it is possible that social desirability bias influenced parental responses. Leading prompts might have also contributed to the positive results.

Finally, the study did not assess the acceptability of remote GMA from the provider's point of view, a necessary consideration along with legislation and infrastructure for future scale-up of the intervention.[14 15]

### CONCLUSION

Engaging parents in the screening of CP with the help of remote GMA is possible in the urban area of a mountainous South Asian lower-middle-income country and is generally welcomed by the participating parents.

**Author affiliations**
[1]Department of Women's and Children's Health, Uppsala University, Uppsala, Sweden
[2]Department of Pediatrics, Gävle Regional Hospital, Region Gävleborg, Gävle, Sweden
[3]Division of children's and women's health, Department of Biomedical and Clinical Sciences, Linköping University, Linköping, Sweden
[4]Neuropediatric Unit, Department of Women's and Children's Health, Karolinska University Hospital, Karolinska Institutet, Stockholm, Sweden
[5]Golden Community, Lalitpur, Nepal
[6]Paropakar Maternity and Women's Hospital, Kathmandu, Nepal
[7]Division of Children's and Women's Health, Department of Biomedical and Clinical Sciences, Linköping University, Linkoping, Sweden
[8]Department of Global Public Health, Karolinska Institutet, Stockholm, Sweden
[9]School of Public Health and Community Medicine, University of Gothenburg, Gothenburg, Sweden

**Acknowledgements** Asmita Basnet recruited the study participants and guided the parents in installing the NeuroMotion application in the hospital. Rabina Karki conducted the phone follow-up, provided support for the parents during filming and administered the usability survey. Anna Axelin contributed to the design of the study. The authors wish to thank Alicia Spittle and Amanda Kwon for the permission to use the usability survey questionnaire.

**Contributors** AJK, AKC, HEKS, HL, JW, KUS, NB, PP and RG designed the study. AJK, AKC, KUS, OB, PB, PP and RG collected the data and/or supervised data acquisition. AJK, HEKS, KS and OB analysed the data. AJK drafted the manuscript and had access to all the data. All authors participated in revising the manuscript critically. All authors approve the submitted version. AKC is the guarantor of the study, had access to the data dn controlled the decision to publish.

**Funding** The project was funded by grants from Stiftelsen Promobilia, 21009, Linnéa and Josef Carlssons stiftelse, Stiftelsen Folke Bernadottes Minnesfond and Födelsefonden. HEKS is funded by Region Stockholm, clinical postdoctoral appointment, 2019-1138, and a private donation through the Knut och Alice Wallenbergs Stiftelse.

**Competing interests** KS and HEKS developed the application NeuroMotion with financial support from a private donation through Knut och Alice Wallenbergs Stiftelse. For the purpose of further research and later implementation of NeuroMotion, the company NeuroMotion AB was founded and is the owner of the application. NeuroMotion AB is owned by KS and HEKS and has the purpose of research and to be non-profitable.

**Patient and public involvement** Patients and/or the public were not involved in the design, or conduct, or reporting, or dissemination plans of this research.

**Patient consent for publication** Not applicable.

**Ethics approval** This study involves human participants and was approved by Institutional Review Committee of the Paropakar Maternity and Women's Hospital (61/1978) and Swedish Ethical Review Authority (2020-07168). Participants gave informed consent to participate in the study before taking part.

**Provenance and peer review** Not commissioned; externally peer reviewed.

**Data availability statement** Data are available upon reasonable request. Data available on request from the authors: The data that support the findings of this study are available from the corresponding author upon reasonable request.

**ORCID iDs**
Antti Juhani Kukka http://orcid.org/0000-0002-5879-7417
Nick Brown http://orcid.org/0000-0003-1789-0436
Ashish KC http://orcid.org/0000-0002-0541-4486

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
