## [Reviewer comments · BMJ Open]

ARTICLE DETAILS

TITLE (PROVISIONAL)	NeuroMotion™ Smartphone Application for Remote General Movements Assessment: a Feasibility Study in Nepal
AUTHORS	Kukka, Antti; Sundelin, Heléne; Basnet, Omkar; Paudel, Prajwal; Subedi, Kalpana; Svensson, Katarina; brown, nick; Litorp, Helena; Gurung, Rejina; Bhattarai, Pratiksha; Wrammert, Johan; Ashish, KC

VERSION 1 – REVIEW

REVIEWER	Kwong, Amanda Murdoch Childrens Research Institute, VIBeS
REVIEW RETURNED	01-Nov-2023

GENERAL COMMENTS	This manuscript details the feasibility study of the Neuromotion app for recording general movements in Nepal. The study demonstrates that using an app is feasible in a LMIC population with good insight into factors needed for successful future scale-up of general movements app use. Some points needing further clarification are outlined below: Introduction: This provides a good, concise background and justification for the need of the current study with appropriate use of references. The specific aims of the study should be clearly outlined at the end of the introduction. Methods: Can the authors please explain why families with a travel time >1 hour to the hospital were excluded, especially given an app would alleviate the issue of travel time to attend appointments? How does the app securely upload videos? Please consider describing the GMA as the Prechtl GMA if this is being used. Statistical analysis needs to be linked with aims presented in the introduction. As the aims are not detailed clearly, the statistical analysis is difficult to comment on. Is there a reference for Statistical analysis plan b) – the standardized GMA trust questionnaire? Statistical plan c) – does not seem to align with the aims of the study. The feasibility should present parent perceptions of the app alone with comparisons made with other studies in the discussion. E.g. this would be more suited to a review study where a systematic process has been undertaken to compare studies with respect to design, population, etc. There are other apps that help record the GMs that are not reflected in this study. Additionally, the previous studies presented to not use the same apps. Results: Technical quality is reported as “acceptable” in 35 out of 46 cases and “good” from 16, but I have missed in the methods what the
---

	threshold is for acceptable technical quality. Supplementary Table 1 is referenced but I am unable to see where the technical quality of the video is reflected. How many videos could be scored using the GMA? While technical quality is interesting, the actual number of videos where a GMA could be scored is important to report. How did the authors determine the comparability of the stacked bar charts? No statistical test is used to compare responses between studies. Additionally, Are there screenshot examples of the app that can be included in this paper? Discussion: This section would be better suited to comment on the parent survey in comparison to other studies where the context and differences of the apps can be discussed. Please review the findings from the Kwong et al. study. For instructions to "replace" an app, the app needs to be in common clinical use. I was not able to find suggestion that instructions should replace apps. Please also review the data transfer methods in references 27 and 28 as secure methods of transfer may have been used, advocated for, and discussed at length in these studies. Limitations and strengths within contextual, cultural considerations are otherwise well described.
--	--

REVIEWER	Peyton, Colleen Northwestern University
REVIEW RETURNED	17-Nov-2023

GENERAL COMMENTS	Thank you for the opportunity to review your paper. Summary: This paper explores the feasibility of using a smartphone app, called Neuromotion to collect and use GMA data in a population of children with birth asphyxia from a single hospital center in Kathmandu Nepal. Strengths: Previous work has demonstrated the feasibility of using smartphone apps to collect GMA data in HIC but has not been shown in LMIC. This study addresses this knowledge gap and is generally well-written and easy to follow. Overall concerns: The largest concern in this paper, as the authors acknowledge, is the small sample size. This may be due to the strict criteria, including only children with birth asphyxia. Perhaps the authors could consider including other children with medical complexity who they would like to follow, perhaps IUGR, low birth weight, infection or other conditions the authors find relevant? Alternatively, the authors could focus on explaining why the population of interest was specifically used for more context. Suggestions: The following suggestions may help to improve the manuscript. Introduction: - In the first sentence, the authors discuss the "right to survive birth". Because the topic of birth survival is not the focus of this paper, I suggest leaving this out and focusing on the developmental potential. As currently stated, this opening sentence does not focus the reader's attention on the main topic of interest.
---

- Adding background information on how early detection guidelines for CP have or have not been implemented in Nepal would be useful to understand why this technology may be useful in this context. Similarly any background on how and when is CP diagnosed and what support services are available in Nepal would also be helpful to know as background.

Study design and setting:

- Currently it is not clear what is meant by "due to a lack of ventilators"-- suggest either clarifying this statement or simply stating that infants who were transferred out of the hospital were excluded.

- What does it mean that "hospital guidelines mandated that all eligible infants be admitted to the NICU/SNCU"? Perhaps you can simplify by instead stating that all included infants were admitted to the NICU/SNCU?

Intervention:

-"The costs of the rehabilitation were covered by the research team." Were there any costs? If not, suggest removing this sentence. In results it seems as if the families who were referred to rehabilitation, declined this.

Results:

"The most common quality issues leading to the rejection of videos were inability to hold the camera still (15/46), wrong position of the camera for filming (12/46) and a background that was too busy (11/46)." Suggest in parentheses writing (n = 15, 32%).

Also- why was a background that was too busy a cause for rejection? While this is not ideal, this can still be scored, correct?

Can you clarify what is meant by the parent who did not return a scorable video--does this mean those who did not return any video and also those who returned a video but it was rejected? It seems like the authors should also compare those who were enrolled but did not return any video to those who returned videos. Perhaps this is what was done and I am misinterpreting? In any case, please clarify.

Discussion:

"Realising the aim of the Sustainable Development Goals agenda of leaving no child behind will only be achieved if remote GMA screening is conducted as part of a follow-up package with simultaneous improvement in pre- and peripartum care." - what is the Sustainable Development Goals agenda? This is the first time it is introduced and does not have a reference.

It would be useful to understand the context of how parents in Nepal would receive and appreciate the earlier diagnosis of CP of a young infant. This may have implications about how guidelines are implemented and how early intervention is provided. There are several studies about this in HIC but less is known in this context. In this small sample the two cases of absent fidgety (who did not move away) declined intervention. The authors suggest that additional qualitative studies are needed but are there any existing references about this or attitudes about disability in this context that could be discussed or referenced in this article?

VERSION 1 – AUTHOR RESPONSE

Reviewer 1:

1. The specific aims of the study should be clearly outlined at the end of the introduction.

Thank you for the suggestion. We added following to the end of Background:

“Feasibility was assessed through following specific objectives:

- 1) to estimate the *reach* of the screening program
- 2) to evaluate the *technical quality* of the videos and *quality of the assessment*
- 3) to survey parental perceptions of the *usability* of the app.”

2. Can the authors please explain why families with a travel time >1 hour to the hospital were excluded, especially given an app would alleviate the issue of travel time to attend appointments?

Thank you for the comment. Access to rehabilitation for children at risk of cerebral palsy is severely limited in Nepal. We had an agreement of co-operation with Self-help Group for Cerebral Palsy in Kathmandu to provide intervention for children that would be identified through the pilot and found it unethical to screen children who would not be able to access intervention through to travel issues.

We revised Methods to:

“those whose parents lived outside of the Kathmandu valley with travel time to the hospital above one hour were excluded due to limited availability of rehabilitation resources outside the capital”

An additional sentence was added to the beginning of Limitation chapter to highlight this shortcoming:

This study was a small single-centre feasibility study conducted in the capital city of Nepal with a relatively high standard of living, education and mobile phone access, and the results are thus not generalizable to rural areas of the country, where remote screening could be most needed.

3. How does the app securely upload videos?

Thank you for the question. Somewhere along the many revisions to the manuscript this important part fell off. Following lines have been added to “Intervention”

Parents were provided with individual log in credentials to the app.

The app automatically uploaded the films to a secure server at Linköping University, Sweden. Apart from the videos and pseudonymized study ID, no other data were transferred.

4. Please consider describing the GMA as the Prechtl GMA if this is being used.

Thank you for the suggestion! We added this specification to Methods:

“All videos were analysed according to Prechtl’s GMA by three independent certified evaluators blinded to the medical history of the patient.[ref 9]”

Reference 9 is to the method book:

Einspieler C, Prechtl HFR, Bos AF, Ferrari F, Cioni G. Prechtl’s Method on the Qualitative Assessment of General Movements in Preterm, Term and Young Infants. Cambridge University Press; 2005. 104 p.

5. Statistical analysis needs to be linked with aims presented in the introduction. As the aims are not detailed clearly, the statistical analysis is difficult to comment on.

Is there a reference for Statistical analysis plan b) – the standardized GMA trust questionnaire?

Thank you for the comment.

We only reported proportions with acceptable technical quality and revised the relevant section b) in Methods to

“The technical quality of the videos produced was evaluated by a single evaluator (HEKS) using standardized GMA Trust questionnaire. Each video was assessed by eight parameters scoring either “Excellent”, “Good” or “Indistinct”. Presence of any single parameter with “Indistinct” score made the video “Not approved” for technical quality (Supplementary Table 1).”

6. Statistical plan c) – does not seem to align with the aims of the study. The feasibility should present parent perceptions of the app alone with comparisons made with other studies in the discussion. E.g. this would be more suited to a review study where a systematic process has been undertaken to compare studies with respect to design, population, etc. There are other apps that help record the GMs that are not reflected in this study. Additionally, the previous studies presented to not use the same apps.

Apologies for misunderstanding! Around the time of submitting the manuscript, I had email exchange with the reviewer regarding extraction of data from their article referenced as 24 in the submitted manuscript. At the time I did not know who would review the manuscript. I later received a clear response from the reviewer with the same advice as above. When the response arrived to me, the manuscript had already been submitted, which I sincerely apologize for. In my email response I promised that “We will only use the approximate figures for discussion as suggested.” My intention was to make the changes after initial decision by the Editor. Following changes have now been made to address the question:

In Methods:

c) Parental perceptions of the usability of the app were evaluated by a multiple-choice questionnaire and the results were presented using stacked bar graphs. [removal of mention about comparison]

In Results:

Removal of sentence: “The answers were comparable to previous studies from high-income settings” and changing Figure 2 (=new Fig 3) into one only presenting our findings.

7. Technical quality is reported as “acceptable” in 35 out of 46 cases and “good” from 16, but I have missed in the methods what the threshold is for acceptable technical quality.

Supplementary Table 1 is referenced but I am unable to see where the technical quality of the video is reflected.

The technical quality of the videos was assessed by eight parameters defined in the Supplementary Table 1. Thanks to your kind comments, we realized that we had been inconsistent in terminology and hope that the change in Methods will clarify the terms. We have tried to stick to “approved” and “not approved” throughout in the revised manuscript.

b) The technical quality of the videos produced was evaluated by a single evaluator (HEKS) using standardized GMA Trust questionnaire. Each video was assessed by eight parameters scoring either “Excellent”, “Good” or “Indistinct”. Presence of any single parameter with “Indistinct” score made the video “Not approved” for technical quality (Supplementary Table 1).

Technical quality was approved in 35/46 videos and 16/20 children had at least one video with approved technical quality. The relevant section in Results reads:

The technical quality was approved in 35 out of 46 videos and 16 children had at least one video with approved technical quality.

8. How many videos could be scored using the GMA? While technical quality is interesting, the actual number of videos where a GMA could be scored is important to report.

Thank you for the important comment. Following sentence was added to Results:

GMA scoring could confidently be performed in four additional videos with not approved technical quality bringing the total number of children assessed by GMA to 18.

9. How did the authors determine the comparability of the stacked bar charts? No statistical test is used to compare responses between studies.

Only a visual determination was made. We propose changing the Figure 2 as discussed in response 6.

10. Additionally, are there screenshot examples of the app that can be included in this paper?

Thank you for the suggestion. We propose adding a new Figure 1 below:

- 11.** This section would be better suited to comment on the parent survey in comparison to other studies where the context and differences of the apps can be discussed.

Thank you for the comment. Please refer to response 6 for our suggested solution.

- 12.** Please review the findings from the Kwong et al. study. For instructions to “replace” an app, the app needs to be in common clinical use. I was not able to find suggestion that instructions should replace apps. Please also review the data transfer methods in references 27 and 28 as secure methods of transfer may have been used, advocated for, and discussed at length in these studies.

Our understanding of the referenced paper was that families were randomized to receive two types of instructions about filming their child at home and uploading the video, but neither group used BabyMoves or another specified “GMA app”. As both groups had high success rate also seen in the Indian study where parents used WhatsApp (ref 28), we deduced that “simple instructions for filming could potentially replace a separate smartphone app for achieving good results with remote GMA.”

As the first author of the referenced study 27 (ref 32 in current manuscript), Dr. Kwong is surely best in place to decide whether instructions could replace a separate app or not. We would kindly request further clarification from the reviewer and editor in case the following revision still does not correctly reflect the contents of study 27 (new ref 32):

“Kwong et al. in Australia recently demonstrated that simple instructions for filming enabled parents to successfully film their children at home for centralized GMA.³² A small pilot study in India did the same using WhatsApp messaging suggesting that a separate app for GMA might not be needed.³³”

- 13.** Please also review the data transfer methods in references 27 and 28 as secure methods of transfer may have been used, advocated for, and discussed at length in these studies.

It was not our intention to imply that data safety was an issue in the referenced studies. We suggest following revision to avoid any potential confusion:

“However, privacy regulations and safety of data transfer and storage need to be considered when implementing telemedicine interventions.¹⁴”

- 14.** Limitations and strengths within contextual, cultural considerations are otherwise well described.

Thank you for the comment.

Reviewer 2:

1. Overall concerns: The largest concern in this paper, as the authors acknowledge, is the small sample size. This may be due to the strict criteria, including only children with birth asphyxia. Perhaps the authors could consider including other children with medical complexity who they would like to follow, perhaps IUGR, low birth weight, infection or other conditions the authors find relevant? Alternatively, the authors could focus on explaining why the population of interest was specifically used for more context.

Thank you for the thoughtful comment. Appropriate selection of a high risk population is important for evaluating a screening intervention like GMA. A much higher proportion of all CP cases in LMICS are due to birth asphyxia than in HICs due to higher incidence of asphyxia and higher mortality among preterm infants. The validity of GMA in this particular population is poorly established.

The present manuscript is a pilot study devised in preparation of a larger scale study that would establish the accuracy of GMA screening for later CP in an LMIC high-risk population. We therefore wanted to include primarily patients that have been underrepresented in previous studies. Based on our experience from the same hospital, we did not expect recruitment issues. Fortunately for the patients and unfortunately for us, the rate of birth asphyxia had decreased since the conduct of the study by KC et al. 2016 (ref 28 in current manuscript).

Following edits were made to address the comment:

In Background:

“It has been shown to have excellent validity at 3 months’ age and its use in screening of high risk children is now globally endorsed, although its accuracy in LMIC populations is yet to be established (new ref 12 to a recent scoping review)”

In Sample Size:

In preparation for this study, we calculated that to show the sensitivity and specificity of GMA for CP diagnosis at two years of age with 95% certainty when the expected sensitivity and specificity are 95% and the expected proportion of infants with abnormal outcome in the population is 20%, then using Buderer’s formula, 365 infants would need to be recruited [27]. A final sample size of 400 would have been needed to account for losses to follow-up. To test the feasibility of recruitment, filming and referral, a convenience sample of 40 neonates was planned for.

2. In the first sentence, the authors discuss the "right to survive birth". Because the topic of birth survival is not the focus of this paper, I suggest leaving this out and focusing on the developmental potential. As currently stated, this opening sentence does not focus the reader's attention on the main topic of interest.

Thank you for the suggestion! We propose following with remaining reference to the WHO Nurturing Care framework for early childhood development:

All children with disability have the right reach their full developmental potential.¹

3. Adding background information on how early detection guidelines for CP have or have not been implemented in Nepal would be useful to understand why this technology may be useful in this context. Similarly any background on how and when is CP diagnosed and what support services are available in Nepal would also be helpful to know as background.

Thanks again for a helpful comment! This matter has been partly discussed under “Study design and setting”. Following additions were made in the same place to provide more context:

“Targeted developmental clinics for high risk children are generally lacking whereby delay in diagnosis until nearly school age is common [24,25]. Paediatric rehabilitation services for CP are provided by non-governmental organizations and government hospitals working by Community Based Rehabilitation principles [26].”

4. Currently it is not clear what is meant by "due to a lack of ventilators"-- suggest either clarifying this statement or simply stating that infants who were transferred out of the hospital were excluded.

The hospital had high load of neonatal infections during the study period whereby many infants were transferred to other hospitals due to temporary lack of ventilators. We followed the reviewer's suggestion and removed the part.

5. What does it mean that "hospital guidelines mandated that all eligible infants be admitted to the NICU/SNCU"? Perhaps you can simplify by instead stating that all included infants were admitted to the NICU/SNCU?

Thank you for the request. We wanted to clarify that all infants fulfilling inclusion criteria were routinely admitted NICU/SNCU as no recruitment was done from the post-natal ward. Following edit has been made:

“All infants potentially eligible for the study were routinely admitted to the NICU/SNCU, which was confirmed by a chart review of 3 preceding months.”

6. "The costs of the rehabilitation were covered by the research team." Were there any costs? If not, suggest removing this sentence. In results it seems as if the families who were referred to rehabilitation, declined this.

There were no costs as all parents refused, which is a major limitation of our study. We find it important to mention that refusal was not related to direct costs of rehabilitation whereby we would like to keep the following sentence:

“The research team offered to cover the direct costs of the rehabilitation.”

7. "The most common quality issues leading to the rejection of videos were inability to hold the camera still (15/46), wrong position of the camera for filming (12/46) and a background that was too busy (11/46)." Suggest in parentheses writing (n = 15, 32%).

Thank you for the suggestion! Revised below:

The most common quality issues leading to rejection of videos were inability to hold the camera still (n=15, 33%), wrong position of camera for filming (n=12, 26%) and too busy background (n=11, 24%).

8. Also- why was a background that was too busy a cause for rejection? While this is not ideal, this can still be scored, correct?

Please kindly see response 8 to reviewer 1.

9. Can you clarify what is meant by the parent who did not return a scorable video--does this mean those who did not return any video and also those who returned a video but it was rejected? It seems like the authors should also compare those who were enrolled but did not return any video to those who returned videos. Perhaps this is what was done and I am misinterpreting? In any case, please clarify.

Thank you for the opportunity to clarify. In Table 1 and Results, the comparison is between parents who returned at least one video with approved technical quality and those who did not. The comparison between those who sent in at least one video of any quality and those who did not send any videos was similar (data not presented). We had been inconsistent with use of words like “acceptable”, “scorable” and “good” in the manuscript as was noted by reviewer 1 in comments 5 and 7 and have revised the section related to your comment into following:

“There were no statistically significant differences in the characteristics between children whose parents succeeded in returning at least one film of approved technical quality versus those who did not (Table 1).”

10. "Realising the aim of the Sustainable Development Goals agenda of leaving no child behind will only be achieved if remote GMA screening is conducted as part of a follow-up package with simultaneous improvement in pre- and peripartum care." - what is the Sustainable Development Goals agenda? This is the first time it is introduced and does not have a reference.

Thank you for the careful reading! Sustainable Development Goals are key for the global child health and disability inclusion whereby we believe they provide important context for our findings. We suggest the following edit including a reference to the SDGs with your kind guidance:

The United Nations' Sustainable Development Goals agenda of leaving no child behind³³ can only be achieved if comprehensive neonatal follow-up is provided simultaneously to improvement in pre- and peripartum care.

11. It would be useful to understand the context of how parents in Nepal would receive and appreciate the earlier diagnosis of CP of a young infant. This may have implications about how guidelines are implemented and how early intervention is provided. There are several studies about this in HIC but less is known in this context. In this small sample the two cases of absent fidgety (who did not move away) declined intervention. The authors suggest that additional qualitative studies are needed but are there any existing references about this or attitudes about disability in this context that could be discussed or referenced in this article?

We are not aware of any papers discussing parental perceptions on early high risk CP diagnosis in LMICs and certainly not in Nepal. Our rehabilitation partners Self-help Group for Cerebral Palsy and another NGO in Kathmandu have conducted some studies in their own patient populations and new references were added in response to question 1 (ref 24 and 25 in the new manuscript). There is also an interesting ethnographic study about pediatric CP conducted in Nepal over 20 years ago (ref 37 in the new manuscript) and several non-referenced papers on general disability inclusion. We chose to stick to the “more research is needed” cliché as none of these papers fills the gap of properly conducted qualitative study on this matter.

In case the reviewers or editor are aware of relevant papers, we would be delighted to revise this section!

VERSION 2 – REVIEW

REVIEWER	Kwong, Amanda Murdoch Childrens Research Institute, VIBeS
REVIEW RETURNED	01-Jan-2024

GENERAL COMMENTS	Thank you to the authors for submitting their response. The manuscript is clearer with only a few minor suggestions remaining: Point 10: Thank you for including screenshots. English translation as a legend/footnote would be helpful here. In response to point 12, perhaps the authors need to revise the word "replace" or give it context as it suggests that a predecessor (a smartphone app) is of no use. Perhaps using the phrase "as a proxy" rather than "replace" is more accurate. Alternatively, in keeping the authors' original words: "Kwong et al. in Australia recently demonstrated that simple instructions for filming could potentially replace a separate smartphone app for achieving good results with remote GMA." to "Kwong et al. in Australia recently demonstrated that simple instructions for filming could be used for achieving good results with remote GMA where no dedicated app is available. However, video file transfer methods would need to comply with local data security and transfer laws."
---

REVIEWER	Peyton, Colleen Northwestern University
REVIEW RETURNED	03-Jan-2024

GENERAL COMMENTS	All comments have been adequately addressed.
--

VERSION 2 – AUTHOR RESPONSE

Reviewer 1:

Point 10: Thank you for including screenshots. English translation as a legend/footnote would be helpful here.

Thank you for the suggestion. We've added the corresponding text used in the English version as a clarification in the figure legends. We leave it to the editor to decide if legend or footnote is a better placement for the text.

In response to point 12, perhaps the authors need to revise the word "replace" or give it context as it suggests that a predecessor (a smartphone app) is of no use. Perhaps using the phrase "as a proxy" rather than "replace" is more accurate. Alternatively, in keeping the authors' original words: "Kwong et al. in Australia recently demonstrated that simple instructions for filming could potentially replace a separate smartphone app for achieving good results with remote GMA." to "Kwong et al. in Australia recently demonstrated that simple instructions for filming could be used for achieving good results

with remote GMA where no dedicated app is available. However, video file transfer methods would need to comply with local data security and transfer laws."

Thank you for the constructive feedback! This section has been revised to:

Kwong et al. in Australia and a small pilot study in India recently demonstrated that simple instructions for filming could be used for achieving good results with remote GMA where no dedicated app is available [32,33]. However, video file transfer methods would need to comply with local data security and transfer laws [15].

Reviewer: 2

All comments have been adequately addressed.

Thank you!